# A Pilot Study Exploring the Impact of a Primary Medication Non-Adherence Intervention among Four Chronic Disease States in One Regional Division of a Large Community Pharmacy Chain

**DOI:** 10.3390/pharmacy11010011

**Published:** 2023-01-06

**Authors:** Danya H. Wilson, Leanne J. Rein, Michele Fountain, Andrea Brookhart, Daniel Atchley, Kenneth C. Hohmeier

**Affiliations:** 1Kroger Health, 2620 Elm Hill Pike, Nashville, TN 37214, USA; 2Department of Clinical Pharmacy and Translational Science, College of Pharmacy, University of Tennessee Health Science Center, 301 S Perimeter Park Dr, Nashville, TN 37211, USA

**Keywords:** adherence, primary medication non-adherence, non-adherence, secondary adherence, community pharmacy, community pharmacist, pharmacist

## Abstract

There is a 12.2% rate of primary medication non-adherence (PMN) among community pharmacy patients. The Pharmacy Quality Alliance (PQA) has developed a standardized definition of PMN to aid stakeholders in addressing PMN. However, little research had been conducted to date on how to address PMN. The objective of the study was to determine the impact of an evidence-based adherence intervention program on PMN rates among four chronic disease states and to identify and characterize factors associated with PMN. Patients at risk of PMN were randomized into a control or intervention group. Those in the intervention group received a live call from a pharmacist to determine reason for and to discuss solutions to overcome PMN. Subjects included adult patients with newly prescribed medications used to treat diabetes, hypertension, hyperlipidemia, and/or chronic obstructive pulmonary disease (COPD). This study occurred in six pharmacies across one regional division of a national supermarket, community pharmacy chain. Prescriptions were considered newly initiated when the same drug, or its generic equivalent, had not been filled during the preceding 180 days. Prescriptions were considered at risk if they had not been obtained by day 7 of it being filled. Prescriptions were considered PMN if the patient had not obtained it, or an appropriate alternative, within 30 days after it was prescribed. During the 4-month intervention period, 203 prescriptions were included in the study with 94 in the intervention group and 109 in the control group. There was a 9% difference (*p* = 0.193) in PMN between the intervention group (44 patients, 47%) and the control group (61 patients, 56%). The therapeutic class most at risk of PMN was statins (34%). Cost (26%) and confusion/miscommunication (15%) were the most common reasons for PMN within the intervention group. Among the four chronic disease states studied, the intervention had the largest impact on hypertension. The PMN intervention did not significantly decrease PMN rates.

## 1. Introduction

Medication non-adherence represents an avoidable barrier to patient care—resulting in more than $100 billion spent on avoidable hospitalizations each year [1,2,3]. There are two subsets that contribute to the public health issue of non-adherence: primary and secondary medication non-adherence. Primary medication non-adherence (PMN) is the failure to fill an initial prescription within an appropriate time frame, while secondary non-adherence is the failure to refill prescriptions within an appropriate time frame [4]. Secondary non-adherence has been a key measure in quality improvement of medication non-adherence; as such, it has been adopted as a measure into the Centers for Medicare and Medicaid Services (CMS) Star Ratings Program. With the focus of non-adherence research on its counterpart, PMN has been identified as a major gap in research [5,6,7]. The Pharmacy Quality Alliance (PQA) is a national quality organization that works to improve medication safety, adherence, and appropriate use through quality measure research [8]. PQA has standardized the definition of PMN in order to capture more meaningful data on the measurement and to guide research to better understand the true occurrence of primary non-adherence rates [8]. By standardizing the definition of PMN, health professionals not only are able to better track the rates of occurrence, but are also able to track the effectiveness of interventions aimed at reducing PMN among patients. Using this PQA measure for PMN, there is a 12.2% rate of primary medication non-adherence (PMN) among community pharmacies [8]. Multiple studies have found that medications prescribed for chronic diseases are among some of the highest rates of PMN [9,10,11]. While different sources give varying definitions of the duration and limitations of a chronic disease, most agree that it is a condition that is long-lasting and requires ongoing medical attention [12,13]. With nearly 50% of the population in the United States having a chronic disease, and 86% of health care costs attributed to chronic disease, it is clear why chronic disease treatment and control is significant [14]. PQA has given priority to medications used to treat diabetes, hypertension, dyslipidemia, and chronic obstructive pulmonary disease (COPD) [8]. These four chronic diseases listed above are among some of the most prevalent and costly to the nation’s healthcare system as a whole [15,16]. Research has also revealed why patients may not pick up newly prescribed medications. Individual patient and medication factors, health system and socioeconomic effects, and provider–patient communication are all contributing aspects to PMN [11]. The objectives of this study are (1) to determine the impact of a pharmacist-led, evidence-based adherence intervention program on primary medication non-adherence (PMN) rates among four chronic disease states (diabetes, hypertension, hyperlipidemia, chronic obstructive pulmonary disease) and (2) to identify and characterize factors associated with PMN.

## 2. Materials and Methods

Select pharmacies across one regional division of a large community pharmacy chain, with varying factors taken into consideration (prescription count, patient populations, and socioeconomics), were chosen to be included in the study. One pharmacist used remote pharmacy dispensing software to identify prescriptions at risk of PMN. Once prescriptions were considered at risk of PMN, they were randomized into a control or intervention group using an Excel randomization function. Those in the control group were followed to determine if the prescription was obtained within the appropriate amount of time. Those in the intervention group were contacted by a pharmacist who implemented an evidence-based protocol to support pharmacist–patient communication. Patients were asked to provide a reason for not obtaining their medication, and in response, the pharmacist implemented various protocols (i.e., if fear of side effects and education was provided).

The study was an exploratory pilot, randomized controlled study using a single-blinded design (patients were blinded to the intervention). It used electronic prescription and internal patient record data to determine the impact of a pharmacist-led, evidence-based adherence intervention program on PMN rates among medications used to treat four chronic disease states. The PQA-PMN measure was used to define PMN. This intervention was implemented across a select number of regional pharmacies of a large community chain over a 4-month period. This study was considered exempt by the University of Tennessee Health Science Center Institutional Review Board.

An a priori sample size power analysis of 88 patients in both the control and intervention groups was calculated with an alpha of 0.05 and an effect size of 0.10. At 2.5 months into the study, it was determined that the rate of enrollment was not enough to meet power by the end of the planned implementation period, so more stores were added to the study. Pharmacy dispensing software was used to identify newly initiated prescriptions within certain therapeutic classes (Table 1) at risk of primary medication non-adherence for patients 18 years of age or older. A newly initiated prescription is defined as the same drug, or its generic equivalent, not being filled during the preceding 180 days [17,18]. A patient was considered “at risk” of PMN if they had not obtained the newly initiated prescription within 7 days of it being filled. A prescription was considered PMN if the patient did not obtain the newly initiated medication, or an appropriate alternative, within 30 days after it was prescribed [17,18]. Prescriptions were included in the study if they were filled any time during the 30 days after being prescribed, (i.e., if the newly initiated medication was not filled until 10 days into the 30-day period, it was still included, giving less time to contact the patient).

If a patient was at risk of PMN, they were randomized into a control or intervention group using a random number generator. The intervention group was contacted via phone by the study pharmacist who used an evidence-based protocol to support pharmacist–patient communication in order to identify barriers and create solutions to overcome potential PMN [19]. The control group was not contacted. The protocol was adapted from the Drug Adherence Work-up (DRAW) Tool and the conversation flowchart used in P. Chancy et al. [19,20]. The DRAW tool has been published as a resource to help reduce non-adherence, and has been primarily used to address secondary non-adherence. The flowchart in P. chancy et al. has been published as a tool for addressing prescription abandonment. This evidence-based tool walks pharmacy staff through screening patients for PMN and creating a conversation about addressing and educating patients on potential PMN. Patients were considered unreachable after three attempts of contact were made.

When the intervention implementation period was over, the rate of PMN was assessed between the control and intervention groups using a chi-square test.

## 3. Results

During the 4-month intervention period (November 2020 through March 2021), 203 prescriptions were included in the study with 94 in the intervention group and 109 in the control group (Table 2). There was a 9% difference (*p* = 0.193) in PMN between the intervention group (44 patients, 47%) and the control group (61 patients, 56%). Most patients were greater than 50 years of age (63%) and male (55%) (Table 3). The therapeutic classes most at risk of PMN include statins (34%), ACE inhibitors (19%), and COPD inhalers (15%) (Table 4). Among the four chronic disease states studied, the intervention had the largest impact on hypertension (Table 5). Cost (26%) and confusion/miscommunication (15%) were the most common reasons for PMN within the intervention group (Figure 1).

## 4. Discussion

The objective of this pilot study was to explore the impact of an evidence-based adherence intervention program in a community pharmacy setting on PMN rates among four chronic disease states and to identify and characterize factors associated with PMN. Although there was a decrease in PMN rates, it was not found to be significant. However, given the exploratory nature of the study, there were several key takeaways from the study which warrant further investigation.

First, the major reasons for PMN included cost, confusion/miscommunication, fear/actual side effects, and forgetfulness. These results confirm previously identified PMN factors found across a multitude of care settings beyond the community pharmacy, but importantly capture reasons from the patient’s perspective—rather than the perspective of the health system broadly [21]. This patient-level perspective will be critical to the development of future studies which aim to target PMN specifically and underscore the need to approach the PMN population differently than those with secondary medication non-adherence. Of particular importance was the finding that patients had difficulty articulating why they had not yet picked up their newly prescribed medication, which may explain the large percentage of “other” factors associated with PMN. This may further underscore the need for pharmacist interventions in this population as patients themselves may be ill-equipped to overcome PMN on their own.

The intervention that was implemented may also be further refined by creating more targeted conversation with patients. Barriers to adherence may be better addressed by creating a PMN intervention tool that targets common factors associated with PMN and having a pharmacy staff member use the tool to better guide conversation. This is a similar approach to the DRAW tool, a resource used to address secondary non-adherence. Another way to improve this intervention would be to better integrate it into pharmacy workflow by automating the identification of patients at risk of PMN [22,23,24]. This enhanced integration into workflow would not only decrease the time it takes to manually identify patients but would also allow a better platform of documentation meaning more pharmacy staff members could be involved in the intervention to improve patient engagement. It has been demonstrated that workflow alerts assist in engaging general adherence discussions with patients within a large pharmacy chain [25]. Lastly, all PMN interventions were delivered over the telephone. The intervention may have benefited from being integrated into workflow during a patient encounter when picking up or dropping off a different prescription medication.

The intervention had the largest impact on prescriptions for medications used for hypertension (ACEI/ARB) which is a similar finding to another study by Fischer et al. [26] This result may be explained by patient perception of the implications of hypertension and the medications used to treat it. When improving the intervention to create more targeted conversations, it may be important to assess patient perceptions of the other types of medications and disease states included in this study.

The majority of PMN literature has focused on determining the occurrence of PMN and what medications, patient populations, and other characteristics are associated with PMN [4,7,9,10,11,21], although there have been some studies that attempted to decrease rates of PMN [26,27,28]. Fischer et al. (2014) used automated reminder calls and live phone calls from pharmacists or pharmacy technicians to engage patients at risk of PMN [27]. The live calls were used to better understand the barriers to medication adherence (patient education, cost interventions, patient motivation, etc.); however, the findings from the conversations during the live calls were not recorded [26]. In this study, automated calls had no effects on PMN, while live calls decreased antihypertensive PMN significantly. Fischer et al. (2015) approached PMN from the primary care office that prescribed the new medication using nurses to reach out to patients; however, the calls were limited to a reminder and potential recording of unprompted reasons as to why a new medication was not obtained. During this study, nurse outreach did not improve primary medication adherence [27]. Hackerson et al. (2018) was a collaboration between a community pharmacy and a primary care office in a pediatric patient population. The collaboration between the pharmacy and the clinic, along with targeted patient-specific interventions, resulted in a significant decrease in PMN [28]. The current study attempted to create a pharmacy-led, targeted intervention that could help create solutions to decrease PMN rates in a community pharmacy setting.

There are important limitations to consider when reading the results of this study. Twenty-seven out of 94 (29%) patients were unreachable within the intervention group. This can be explained by certain barriers that present due to cold calling. First, even with caller identification, patients may not recognize their local pharmacy’s number and be unwilling to answer. Second, due to the nature of the phone system used, voicemails could not be left to explain the reason for calling and to request a call back. An additional limitation was that claims data from only one large community pharmacy chain was used. This means that patients could have been filling a medication at another pharmacy and then switched to one of the pharmacies included in the study, making their prescription look like a newly prescribed medication when it truly was not. The PQA-PMN measure includes medication classes that are typically used for four certain chronic diseases, but certain medications included in the study can be used for diseases other than the included four (i.e., metformin for polycystic ovary syndrome [PCOS]). The PQA-PMN measure directs the user to only include electronically prescribed prescriptions; however, this study did not differentiate between electronically prescribed and written prescriptions due to the nature of the pharmacy dispensing software and the way the prescriptions were screened for inclusion. Ultimately, it is not believed that this made a large impact on the study overall [29,30].

## 5. Conclusions

In conclusion, while there was an increase in adherence rates after the implementation of a pharmacist-led PMN intervention, it was found to be not significant. Factors such as more targeted conversations and enhanced integration of protocol into workflow should be further investigated as facilitators to improving PMN via a pharmacy-based intervention.

## Figures and Tables

**Figure 1 pharmacy-11-00011-f001:**
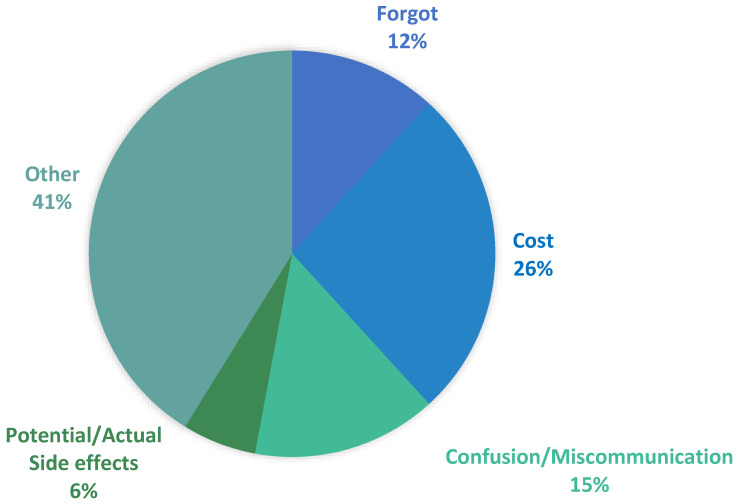
Factors associated with PMN in intervention Group *. * 11 patients were unreachable.

**Table 1 pharmacy-11-00011-t001:** Therapeutic classes included in PQA-PMN measure [8,17,18].

Therapeutic Classes
Angiotensin-converting enzyme (ACE) inhibitors, plus combination products
Angiotensin II receptor blockers (ARBs), plus combination products
Biguanides, plus combination products
Chronic obstructive pulmonary disease (COPD) medications
Direct renin inhibitors, plus combination products
Dipeptidyl peptidase 4 (DPP-IV) inhibitors, plus combination products
Hydroxymethlglutaryl-CoA (HMG-CoA) reductase inhibitors, plus combination products
Incretin mimetic agents
Inhaled corticosteroids
Meglitinides, plus combination products
Sulfonylureas, plus combination products
Thiazolidinediones, plus combination products
Sodium-glucose co-transporter type 2 (SGLT2) inhibitors

**Table 2 pharmacy-11-00011-t002:** Rate of primary medication non-adherence (PMN).

Intervention (*n* = 94)	Control (*n* = 109)	*p* Value
44/94 (47%)	61/109 (56%)	0.193

**Table 3 pharmacy-11-00011-t003:** Patient characteristics.

Characteristic	All Patients (*n* = 203)	Intervention (*n* = 94)		Control (*n* = 109)	
		All	PMN (*n* = 44)	All	PMN (*n* = 61)
Age					
18–24	5 (2%)	2 (2%)	1 (2%)	3 (3%)	3 (4.9%)
25–39	24 (12%)	11 (12%)	3 (7%)	13 (12%)	6 (9.8%)
40–49	47 (23%)	28 (30%)	11 (25%)	19 (17%)	11 (18%)
50–64	81 (40%)	32 (34%)	15 (34%)	49 (45%)	29 (47.6%)
65+	46 (23%)	21 (22%)	14 (32%)	25 (23%)	12 (19.7%)
Sex					
Male	112 (55%)	54 (57%)	29 (66%)	58 (53%)	34 (56%)
Female	91 (45%)	40 (43%)	15 (34%)	51 (47%)	27 (44%)

**Table 4 pharmacy-11-00011-t004:** Drug classes associated with PMN risk *.

Medication Class	*n* = 203
ACE inhibitor + combos	39 (19%)
ARB + combos	21 (10%)
Biguanides	20 (10%)
Biguanides + combos	2 (1%)
COPD inhalers	30 (15%)
DPP-4 inhibitors	4 (2%)
DPP-4 inhibitor/SGLT-2 inhibitor	1 (1%)
Incretin mimetic agents	6 (3%)
Meglitinides	2 (1%)
SGLT2 inhibitors	5 (2%)
Statin medications	68 (34%)
Sulfonylureas	5 (2%)

* risk is defined as newly prescribed prescriptions not obtained by patient within 7 days of fill date.

**Table 5 pharmacy-11-00011-t005:** Intervention subgroup analysis by disease state.

Disease State	Intervention Adherent (*n* = 50)	Intervention PMN (*n* = 44)	Intervention All (*n* = 94)
COPD	5	5	10
Diabetes	12	14	26
Hyperlipidemia	16	17	33
Hypertension	17	8	25

## Data Availability

The data presented in this study are available on request from the corresponding author. The data are not publicly available due to privacy restrictions.

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
