# Peer review of "A Pilot Study Exploring the Impact of a Primary Medication Non-Adherence Intervention among Four Chronic Disease States in One Regional Division of a Large Community Pharmacy Chain"

_pharmacy, 2023, doi:10.3390/pharmacy11010011_

Round 1

Reviewer 1 Report

The manuscript was clear and sound in its presentation of the defined problem and the method used  by the authors to resolve the patient non-adherence, which is widely known problem associated with the drug prescriptions. The authors have used pharmacist-intervened method to reduce the non-adherence, which showed a non-significant improvement in the non-adherence pattern, which is reflected in the issues given in the results and discussion. Overall, the manuscript is well written, informative to the feild and can be considered for the publication.

Author Response

Thank you for your comments. I appreciate the feedback.

Reviewer 2 Report

The manuscript entitled “A Pilot Study Exploring the Impact of a Primary Medication Non-Adherence Intervention Among Four Chronic Disease States in One Regional Division of a Large Community Pharmacy Chain’’ submitted by Wilson DH et al, discusses about the rate of primary medication non-adherence (PMN) among community pharmacy patients and also mentions how to address PMN.

 Authors also mention that this study occurred in six pharmacies across one regional division of a national supermarket, community pharmacy chain. In that prescriptions are considered newly initiated when the same drug, or its generic equivalent, had not been filled during the preceding 180 days. And the prescriptions were considered at risk if they had not been obtained by day 7 of it being filled. Prescriptions were considered PMN if the patient had not obtained it, or an appropriate alternative, within 30 days after it was prescribed.

During the 4-month intervention period, 203 prescriptions were included in the study with 94 in the intervention group and 109 in the control group. There was a 9% difference (p=0.193) in PMN between the intervention group (44 patients, 47%) and the control group (61 patients, 56%).

Authors also look back at the literature and the history for the lessons to take from previous studies and conclude by discussing the challenges for the implementation of a successful future technology.

Overall, I believe adding few more sections mentioned in the general comments to authors will add a comprehensive context to persue the further research on the PMN and provide new exciting areas in expanding its horizon in context to the therapeutic intervention.

General comments:

Ø  I wonder if authors would shed some light and discuss, Factors Associated with PMN in Intervention Group.

Ø  Authors have mentioned Intervention Subgroup Analysis by Disease State. I was wondering if the authors would discuss                                                                                                   in brief       these disease states.

Ø   Rather than making a table of Drug Classes Associated with PMN Risk, I would appreciate if the authors would discuss the drugs and selective Inhibitors of PMN.

Ø  I would like to mention here that it would look more better if the authors discuss this pathway in more detail. As there are some nice studies from various groups using either patient data describing these and the use of drugs and inhibitors should be discussed.

Author Response

Thank you for your comments and feedback, they are very much appreciated.

  1. The factors associated with PMN are simply the reasons patients gave for not picking up the medication in question. 
  2. The disease states are mentioned in table 5 and are described as chronic diseases. Please see the revised manuscript lines 72-75 for additional comments on the chronic disease states. 
  3. The table of drug classes is in response to the PMN measure from PQA that was used. It is what best describes the way we determine which medications to include. 
  4. I am uncertain what pathway is suggested to be elaborated on so I am unsure how to respond to this suggestion.

Reviewer 3 Report

The study described in this paper by the authors is really significant and explore the patient non-adherence to the drug therapies and what are the real reasons that push them to "forget" drugs.

The applied method is rigorous and I recommend it for publication in this journal.

Author Response

Thank you for your comments and feedback.